# Gene drive designs for efficient and localisable population suppression using Y-linked editors

**René Geci** [ID][¤], **Katie Willis** [ID], **Austin Burt** [ID]*

Dept. of Life Sciences, Imperial College London, Silwood Park, United Kingdom

¤ Current address: Institute for Systems Medicine with focus on organ interactions, University Hospital RWTH Aachen, Aachen, Germany
* a.burt@imperial.ac.uk

**Data Availability Statement:** Model code and simulation results are freely available on GitHub (https://github.com/ReneGeci/LocalisableYLEsuppression).

## Abstract

The sterile insect technique (SIT) has been successful in controlling some pest species but is not practicable for many others due to the large number of individuals that need to be reared and released. Previous computer modelling has demonstrated that the release of males carrying a Y-linked editor that kills or sterilises female descendants could be orders of magnitude more efficient than SIT while still remaining spatially restricted, particularly if combined with an autosomal sex distorter. In principle, further gains in efficiency could be achieved by using a self-propagating double drive design, in which each of the two components (the Y-linked editor and the sex ratio distorter) boosted the transmission of the other. To better understand the expected dynamics and impact of releasing constructs of this new design we have analysed a deterministic population genetic and population dynamic model. Our modelling demonstrates that this design can suppress a population from very low release rates, with no invasion threshold. Importantly, the design can work even if homing rates are low and sex chromosomes are silenced at meiosis, potentially expanding the range of species amenable to such control. Moreover, the predicted dynamics and impacts can be exquisitely sensitive to relatively small (e.g., 25%) changes in allele frequencies in the target population, which could be exploited for sequence-based population targeting. Analysis of published *Anopheles gambiae* genome sequences indicates that even for weakly differentiated populations with an $F_{ST}$ of 0.02 there may be thousands of suitably differentiated genomic sites that could be used to restrict the spread and impact of a release. Our proposed design, which extends an already promising development pathway based on Y-linked editors, is therefore a potentially useful addition to the menu of options for genetic biocontrol.

## Author summary

Some pest populations can be successfully controlled by the inundative release of sterile males, but this approach is not practicable when the target population is large or the species difficult to rear. Computer modelling has previously demonstrated that releasing

**Funding:** Supported by a grant from the Bill & Melinda Gates Foundation (Grant INV006610 "Target Malaria Phase II", to AB) and the Open Philanthropy Project Fund, an advised fund of Silicon Valley Community Foundation (Grant O-77157 to AB). The funders had no role in study design, data collection and analysis, decision to publish, or preparation of the manuscript.

**Competing interests:** The authors have declared that no competing interests exist.

males with a genomic editor on their Y chromosome that kills or sterilises female descendants could be much more efficient, particularly if combined with a sex ratio distorter. Here we extend this work to show that Y-linked editors can also be used in even more efficient gene drive designs that would spread over successive generations beyond the region of release. Such spread could nonetheless be controlled by exploiting relatively small pre-existing differences in gene frequency between populations to restrict the spread and impact of the constructs, if desired. The proposed design does not require high rates of recombinational repair of DNA breaks or expression off the Y chromosome during meiosis, potentially expanding the range of species in which such low release rate control is possible. Y-linked editors may therefore form the basis of a highly flexible set of genetic strategies for population control.

## Introduction

The sterile insect technique (SIT) of pest control involves the mass rearing, sterilisation, and release of insects that will mate with members of a target population and thereby interfere with their reproduction and reduce their numbers in the next generation [1]. The approach has been used with notable success against a number of agricultural pests, including screwworm, fruit flies, pink bollworm and others [2]. Sterility has traditionally been achieved by irradiation, but there are closely related approaches using endosymbionts like *Wolbachia* or transgenes conferring dominant lethality that are being explored in mosquitoes and tsetse flies [3]. It is species-specific in its impacts and becomes more efficient as the target population is suppressed, but rearing and releasing a sufficient number of sufficiently vigorous sterile males to ensure that they mate the majority of females in the target population may not be possible if the insect is difficult to rear or the target population very large, limiting the range of use cases for this otherwise attractive technology.

This requirement for mass releases arises because the sterilising property of the released males disappears when those males die, and every generation a fresh set of sterile males needs to be released. In principle, if a system could be devised in which females are genetically sterilised (or killed), but the genes responsible did not disappear, it could be much more efficient. Burt and Deredec [4] propose and analyse one way to achieve this objective, which is to use genome editors that are located on the Y chromosome and act in the male germline to make dominant edits on the X chromosome causing those that inherit the X (i.e., daughters) to die or be sterile. The system can also work if the Y-linked editor (YLE) makes dominant edits at an autosomal locus that affect females but not males. In either case the YLE is not itself selected against due to the mutations it creates, because the Y is not found in those females. Thus repeated releases in successive generations can lead to a gradual increase in the frequency of the YLE and the reproductive load imposed on the population, rather than starting from zero each generation, as is the case with sterile males. Computer modelling suggested that this approach could be much more efficient than sterile male releases, requiring perhaps 10-fold fewer males to be released. Moreover, the modelling also showed that further efficiencies could be gained by releasing the YLE along with an autosomal sex distorter (ASD) that favours transmission of the Y (e.g., by shredding the X chromosome during male meiosis [5]). When released together in the same males, the ASD boosts the transmission rate of the YLE into the next generation, mimicking the effect of a larger release. The ASD itself does not drive, instead being inherited in a Mendelian manner, and would gradually disappear from the population, and so the impacts would still be self-limiting. With this combined system a single release

equivalent to less than 10% of the target population may be enough to substantially suppress it, about 100x more efficient than SIT.

These efficiency gains over traditional SIT would be sufficient for many use cases, but in others the release rates required may still be beyond what is feasible in some species, particularly those that are especially difficult to rear and where the target population is particularly large. These cases may require an even more efficient approach. Putting a sex distorter directly onto the Y chromosome, to create a Driving Y, could, in principle, result in a very efficient control system requiring only trivially small releases [6–8]. Instead of disrupting the survival of progeny inheriting the X, as a YLE would, such a construct would disrupt the transmission of the X chromosome, thereby favouring its own transmission. All else being equal, the Driving Y would increase in frequency over successive generations, gradually making the sex ratio more and more male biased, and thereby suppressing or even eliminating the population. However, while there are many natural examples of driving sex chromosomes [9,10], including driving male-determining regions in two species of mosquito [11], and sex chromosome distorters have been engineered that work if inserted on an autosome [5,12–14], synthetic Driving Ys are thus far proving difficult to engineer, due at least in part to the silencing of the sex chromosomes at male meiosis seen in many species [15].

In this paper we return to the idea of having separate YLE and ASD constructs, and explore the consequences of making a small addition to the autosomal construct that allows not only the ASD to increase the transmission of the YLE, but also for the YLE to increase the transmission of the ASD, to make a "double drive" [16]. In particular, we consider a design in which a gRNA gene is added to the ASD that allows it to home in the presence of the YLE. As will be seen, this small alteration in the design changes the dynamics considerably, so that trivially small release rates can lead to spread of the constructs and suppression of the population, similar to a Driving Y. Importantly, the approach does not rely on sex chromosome expression during meiosis, and can work even with low homing rates, thereby potentially expanding the range of species for which low-release-rate population suppression gene drive approaches are feasible. Moreover, we show that by appropriate design, relatively small pre-existing differences in allele frequencies at a polymorphic site between target and non-target populations of the same species (or species complex) can be used to restrict the spread and impact of the drives (though if control is desired throughout a species range, that would be possible too). We then analyse published genome sequence data from *Anopheles gambiae* s.l. mosquitoes which suggests even weakly differentiated populations could, in principle, be differentially targeted with appropriately designed constructs. These results demonstrate the potential flexibility of YLEs for population suppression, with an incremental step-by-step development pathway of increasingly efficacious constructs, from self-limiting genome editors to self-sustaining double drives, each step building molecularly on the one before.

## Methods

### Genetic design

The design under consideration is an extension of those proposed and analysed by Burt and Deredec [4]. It consists of two constructs, one on the Y chromosome and one on an autosome. A number of different molecular constructions could be used, but for concreteness we will assume a specific configuration (Fig 1). The Y-linked construct encodes a genome editor that targets a gene on the X-chromosome, and the edits induced cause dominant lethality or sterility in the progeny. Because the progeny of a male that inherit his X chromosome are female, it is his daughters that are killed or made sterile. In the specific example configuration, the editor consists of a Cas9 nuclease gene with control sequences that mean it gets expressed in the male

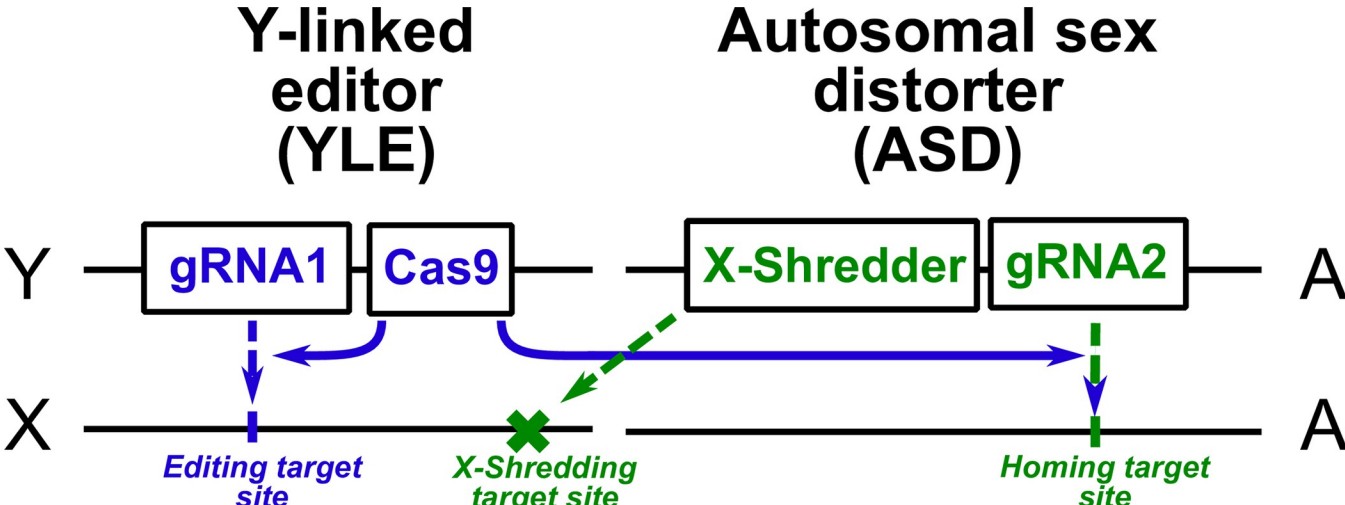

**Fig 1. The proposed genetic constructs.** We model a Y-linked editor (YLE) that targets an X-linked locus such that female offspring die and an autosomal sex distorter (ASD) that in males disrupts transmission of the X chromosome (thereby favouring transmission of the Y), and that, in the presence of the YLE, shows drive via the homing reaction. More specifically, we model the YLE as consisting of a germline-expressed Cas9 and ubiquitously expressed gRNA (together responsible for the editing), and the ASD as consisting of a X-shredder nuclease expressed during male meiosis (responsible for the sex distortion) and a ubiquitously expressed gRNA that, in the presence of the Y-linked Cas9, allows the ASD to home.

germline, and a ubiquitously expressed gRNA gene targeting the X-linked target gene. The autosomal construct encodes a nuclease that targets a repeated sequence on the X chromosome (separate from the target of the YLE), thereby shredding the X, and is expressed during male meiosis, so that males with the construct produce mostly Y-bearing sperm and male offspring [5]. It also carries a ubiquitously expressed gRNA gene targeting its insertion site, so that in heterozygous males, in the presence of the Y-linked construct, the autosomal construct can home in the germline and thereby be transmitted at a greater than Mendelian rate to the offspring. It is the addition of this autosomal gRNA gene that distinguishes this design from the "augmented YLE" design modelled by Burt and Deredec [4] and the autosomal sex distorter strains published by Galizi et al [5].

Note that we now have a double drive [16] in which the transmission of each construct depends in some way on the presence of the other, though in somewhat different ways. The X-shredder causes whatever Y chromosome it is with to show superMendelian transmission, and if there is a statistical association (i.e., linkage disequilibrium) between the two constructs, then the YLE will increase in frequency due to the presence of the X-shredder. At the same time, the autosomal construct can show superMendelian transmission via the homing reaction, but only in the presence of the Y-linked construct. Note also that the editor function *per se* is not directly involved in the drive of either construct, and even if the gRNA responsible for the editing was absent the rest would act as a double drive and lead to some suppression of the target population due to the sex ratio distortion produced by the X-shredder. However, as we will see, including the editor function makes a key quantitative contribution to the function of the whole in terms of increasing the magnitude and duration of suppression.

## Modelling

We use a deterministic population genetic and population dynamic model based on one used previously [4,16] to investigate the fate of the proposed genetic constructs and their impacts on population size. In brief, the model assumes non-overlapping generations and two life

stages, juveniles and adults, with density-dependent mortality occurring during the juvenile phase. Mating is random, and all females are assumed to mate (i.e., males are not limiting). The constructs are introduced into a target population by release of heterozygous males equivalent to 0.1% of the target population.

The model allows for both intended and unintended fitness effects. The intended fitness effects include the effect of the edited allele on female fitness (our baseline assumption is it causes fully dominant and penetrant lethality of adult females, after density-dependent mortality and before censusing) and the fitness effects that arise automatically from distortions of the sex ratio [17]. Unintended fitness costs include those due to expression of the proteins and gRNAs (e.g., the energetic costs of their synthesis) and potential costs due to the activity of the various editors and nucleases (e.g., due to off-target events elsewhere in the genome).

Finally, our model considers a variety of types of mutation. Each component of each construct can be affected by loss-of-function mutations occurring at a constant background mutation rate. Loss-of-function mutations can also occur on the autosomal construct at an elevated rate during homing. Finally, the processes of editing, shredding, and homing can lead to the production of functional resistant sequences at each of the three target sites (e.g., through end joining repair). For simplicity we make the conservative assumption that the resistant sequences have the same fitness as the wild-type alleles. More details about the mechanics of the model and a table of parameters and their baseline values are given in the S1 File. Model code and simulation results are freely available on GitHub (https://github.com/ReneGeci/LocalisableYLEsuppression).

### *An. gambiae* PAM site analysis

To better understand the feasibility of differentially targeting populations of the same species we analysed published genomic sequence from 15 populations of the malaria mosquito *An. gambiae* [18]. The most commonly used CRISPR-Cas9 system requires an NGG sequence in the protospacer adjacent motif (PAM), so we screened for polymorphic GG (or CC) dinucleotide sites and for each pair of populations scored the number of sites that showed the appropriate difference in allele frequencies as determined by the modelling. Further details are given in S1 File.

## Results

### Idealised case

We first consider the simplest idealised case where there are no unintended fitness costs, loss-of-function mutations, or functional or non-functional resistance, and assume molecular efficiencies (editing, homing, X-shredding rates) are high but not perfect, within the range seen in *An. gambiae* mosquitoes [90–95%; 5, 19, 20–22]. In this case if males carrying the two constructs are released, then both the YLE and the ASD constructs increase to and remain at a very high frequency (Fig 2A). The YLE increases in frequency because of the statistical association between it and the ASD. Initially, this correlation is 1 (because they are released in the same males), but gradually it declines because of less than perfect homing and X-shredding rates. The ASD increases in frequency because it homes in the presence of the YLE. It does not home in females or in wildtype males and spreads more slowly than the YLE. As a result of the joint action of the YLE causing dominant female sterile mutations and the ASD producing a male-biased sex ratio, the reproductive load imposed on the population (i.e., the proportionate reduction in reproductive output) rises to 0.99, and the number of females (and therefore total population size) crashes in about 20 generations.

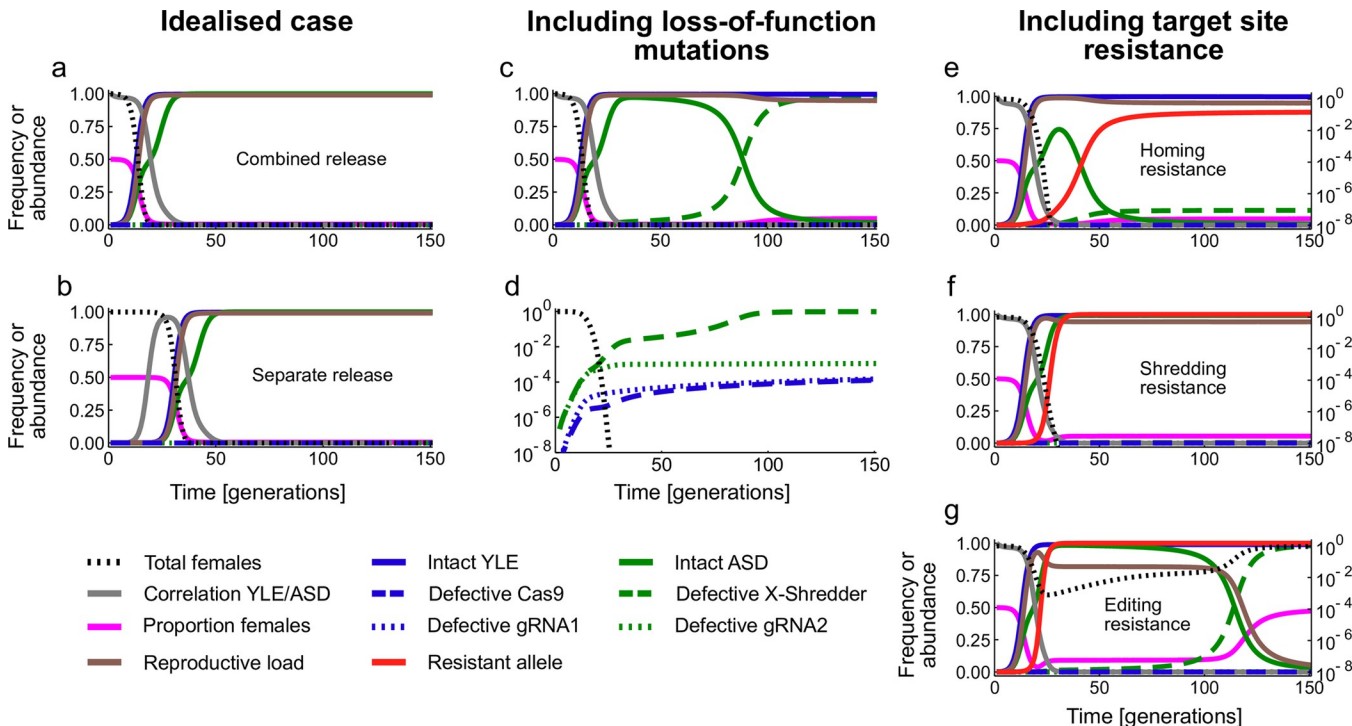

**Fig 2. Timecourse of gene and population dynamics.** (a, b) Idealised case (no mutation, no resistance, no unintended fitness costs) with release of constructs in the same males (a) or in separate males (b). (c, d) Allowing for loss-of-function mutations in the constructs, with the frequency of the most common alleles on an arithmetic scale (c) and frequency of defective variants on a log scale (d). (e-g) Allowing for end-joining repair to produce resistant alleles at the site of homing (e), shredding (f) or editing (g). Allele frequencies are shown on the arithmetic scale (left) and population size (number of females) on the log scale (right). All outputs are calculated at the adult stage, after female mortality due to the edited allele, and population sizes are relative to the pre-release equilibrium.

To better understand the processes underlying these dynamics, we analysed the consequences of releasing the two constructs in different males (Fig 2B). Though the spread of the YLE depends on a statistical correlation with the ASD, it is not necessary for the two constructs to be released in the same males for them to spread. If they are released in different males (in which case the correlation between them is initially negative), the fact that the YLE helps the ASD to drive means that a positive correlation gradually builds up, allowing the two constructs to spread, before the correlation falls again. The dynamics of spread and impact on the population are much the same as with combined releases, just with a delay. On the other hand, neither construct, if released by itself, increases in frequency (S1 Fig). This ability to spread through and crash a population from very small releases is what distinguishes this double drive design from previously proposed YLE strategies, whether used alone or with a non-driving ASD (S2 Fig).

## Including loss-of-function mutations

In the idealised model, we just considered there are two types of Y (wildtype or transgenic), two types of X (wildtype or edited), and two types of autosome (wildtype or transgenic). We now consider the more realistic case in which mutations can arise in the various components of the two constructs which abolishes their function. We assume the loss-of-function mutations occur in the two components of the YLE (the Cas9 and the gRNA) and the two components of the ASD (the X-shredder and the gRNA) with a probability of 1e-6 per generation, due to errors of normal DNA replication, and, moreover, that mutations arise in the two

components of the autosomal construct with a probability of 1e-3 per homing event, as homing is likely to be associated with a larger mutation rate than normal DNA replication [23–25].

The dynamics in this case are initially the same as in the idealised case, with the two constructs increasing in frequency, and population size declining (Fig 2C). But then a defective autosomal construct that has the gRNA but is missing the X-shredder begins to accumulate, replacing the intact construct. These defective elements arise predominantly during the homing reaction, and they rapidly increase in frequency because they are still able to home and yet are associated with a 50:50 sex ratio, which is strongly selected for in a population that has become strongly male-biased. Moreover, the correlation between the YLE and ASD is small by this time, so X-shredding has less of an effect on the probability of homing in the next generation than immediately after the release. As a result, the proportion of adults that are female rises slightly (though is still substantially depressed by the mortality caused by the edited allele), and the reproductive load dips slightly. Nevertheless, the population is still very substantially suppressed because, before being lost, the X-shredder boosts the YLE to a very high frequency, and the YLE is the main source of the reproductive load. While the X-shredder does contribute directly to the load, its main function is indirect, via boosting the frequency of the YLE.

Defective variants of the other three components of the original constructs do not have the same dynamics and remain rare (Fig 2D). The gRNA in the autosomal construct allows it to home, and so loss-of-function mutations in it are selected against as long as there are still sites to home in to. The Cas9 in the YLE helps maintain the association with the X-shredder, and so loss of that too will be selected against. Finally, the gRNA in the YLE does not help the YLE spread, but the pressure on it to be lost is weak. Since this construct does not home, the only source of loss-of-function mutations is normal DNA replication, and these arise rarely (1e-6 in our model) and accumulate slowly. If there are fitness costs associated with any of these functions (e.g., due to costs of gRNA or Cas9 synthesis, or off-target cleavage), then these might accelerate the loss of one or other of these functions, but the effect will be modest unless the costs are large.

## Target site resistance

The three molecular processes in our design (editing, shredding, and homing) each occur at a different target site, and each one could give rise to a resistant sequence that is no longer recognised by the enzymes involved (e.g., by end-joining repair, EJR). We now consider the consequences of allowing such mutations to arise, assuming the most conservative case where resistant alleles have the same fitness as the wildtype allele.

**Homing resistance.**   Our baseline assumption is that the insertion has no fitness cost, and the simplest way to achieve this would be to insert it in a neutral non-functional part of the genome, in which case all cleavage-resistant sequences are as "functional" as the wildtype, and the frequency of functional resistance mutations is then just the frequency of end-joining repair (plus any partial homing events producing wholly defective inserts). The effect of including even a small frequency of EJR (5%) at the autosomal locus is that it is the resistant allele that accumulates rather than the defective X-shredder, but still there is only a slight dip in the reproductive load (Fig 2E). Indeed, we still observe strong suppression even with EJR rates up to 75% (Fig 3A). Again, the spread of the ASD is not directly responsible for suppressing the population–it merely functions to boost the YLE to a high frequency, and it does not have to reach a high frequency itself to achieve that. This ability of the design to function even in species with weak and inefficient homing rates could be an attractive feature, expanding the range of species in which such control is possible, and, because of the potential interest, all

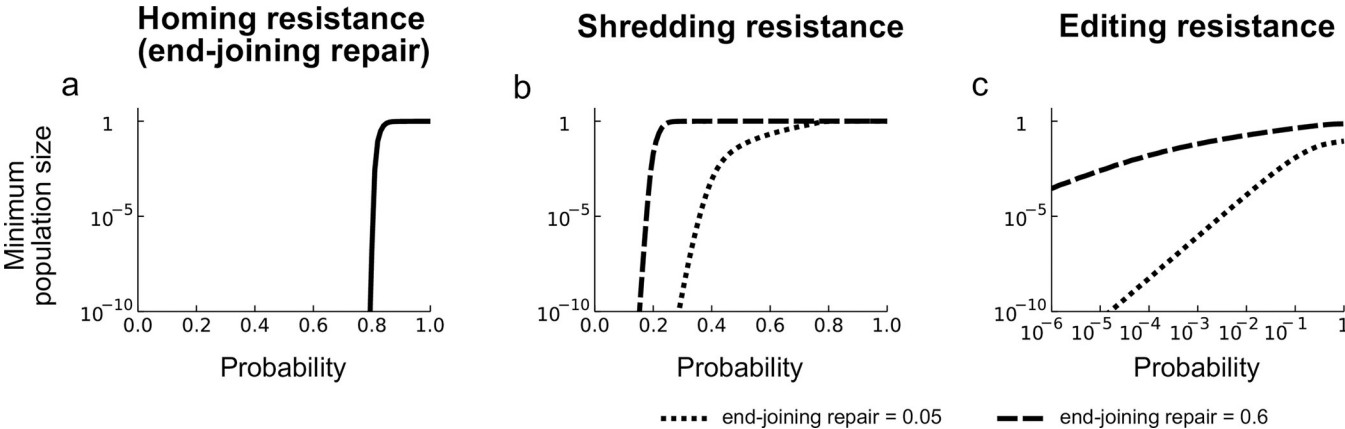

**Fig 3. The impact of target site resistance.** Minimum population size (relative to the pre-release equilibrium) as a function of the probability of a resistant allele forming at the site of homing (a), shredding (b), and editing (c; note log X-axis). For (b) and (c) results are shown for both low and high EJR scenarios (5% and 60% EJR, respectively).

subsequent modelling in this paper will be done for both low (5%) and high (60%) EJR scenarios or species.

**Shredding resistance.**   The X-shredder is likely to need to target a repeated sequence on the X, and so it may be that resistance would require simultaneous changes at many sequences and may therefore be unlikely to arise. Nevertheless, we can still assess the consequences if it did. If we assume it arises in 5% of shredding events, then the modelling again shows that resistance evolves quickly but the population is still very substantially suppressed (Fig 2F). Again, it is not the shredding that is primarily responsible for the load on the population, but the editing. Even higher probabilities of shredding resistance arising can still lead to substantial suppression, even in high EJR species (Fig 3B). Note that if shredding resistance evolves then there is no selection in favour of the defective X-shredder element, because shredding is not occurring anyway (Fig 2F).

**Editing resistance.**   Finally, if we allow for editing resistance at 5%, then, again, resistance evolves very quickly, but now the reproductive load decreases more substantially and the population is suppressed to a far lesser extent (minimum population size 8e-4 relative to the original, after which it recovers; Fig 2G). Moreover, while for both homing and shredding resistance there is a relatively rapid transition between very strong suppression and virtually none as the probabilities of resistance arising change, for editing resistance the picture is somewhat different (Fig 3C). As noted above, even if there is no editing at all, the double drive can lead to some level of suppression due to the sex-distorting effect of the X-shredder. However, most of the load imposed on a population is due to the editor, and if functional resistance does arise at the target of editing it is rapidly selected for. As a result, even low rates of resistance (e.g., 1e-3) significantly reduce the level of suppression. This is particularly the case in high EJR species; the strategy is somewhat more resilient in low EJR species. Nevertheless, the modelling suggests that most effort and attention in preventing functional resistance should go into the design of the editing function.

## Sensitivity analysis for other parameters

The model includes a number of other parameters, and the impact of varying these (while leaving the others at their baseline values; Table A in S1 File) is shown in Fig 4, for both low and high EJR scenarios. The effect of varying the efficiencies of the three processes is as expected

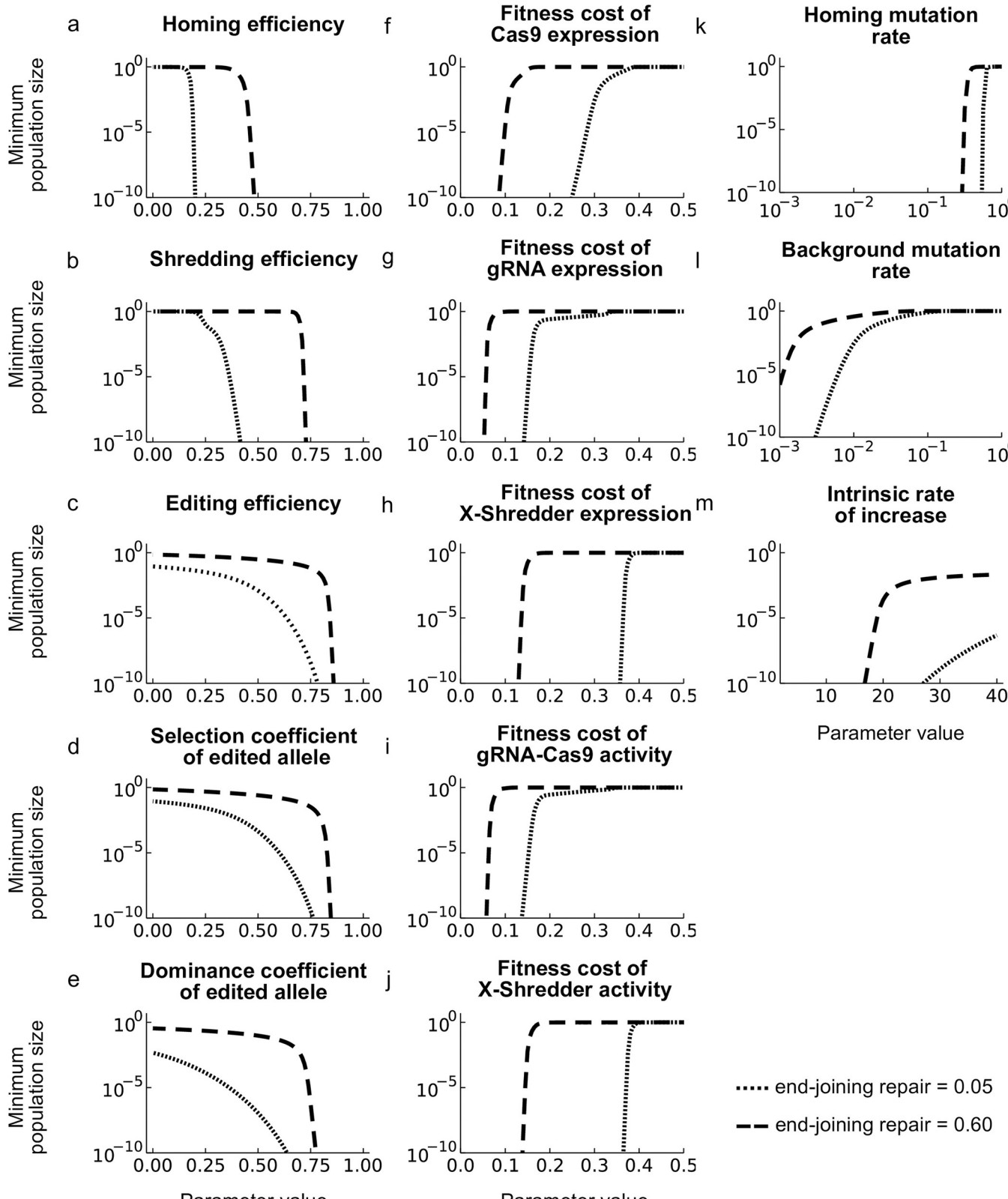

**Fig 4. Sensitivity analysis.** Each plot shows the minimum population size as a function of changes in the specified parameter assuming low EJR (5%; dotted lines) or high (60%; dashed lines). Note for (a) that the homing rate is equal to (cleavage rate) x (1-end joining rate).

from the resistance analysis: the requirements on homing efficiencies are the least stringent, followed by the shredding efficiency, and finally the requirement on editing efficiency is the most stringent, with rates needing to be greater than about 80% in high EJR species to have a strong suppressive effect (Fig 4A–4C). Note again that if editing efficiency is 0 there can still be some suppression, due to the sex ratio distortion produced by the X-shredder, but it is relatively small (Fig 4C). The effect of the edit on female fitness must be similarly high and dominant (Fig 4D–4E).

Six unintended fitness costs are modelled. Under our baseline assumptions the fitness cost of the edited allele in males does not matter because the edited allele never occurs in males: it is created on the X-chromosome in the male germline, and then transmitted to daughters, who die with 100% probability. If the target of editing was autosomal, then fitness effects in males would matter more [4]. For the other five unintended fitness costs there is a relatively sharp threshold between very good suppression and virtually none somewhere between fitness costs of about 0.15 and 0.35 (low EJR), or 0.05 and 0.15 (high EJR) (Fig 4F–4J). The requirements on costs of gRNA expression and of Cas9-gRNA activity are the most stringent, presumably because each of them is doubled due to there being two gRNAs.

Population suppression is robust to reasonable values for loss-of-function mutation rates. For homing-associated mutations, rates up to about 50% are consistent with strong suppression, and for normal cell-division-associated mutations rates of up to 1e-3 (Fig 4K–4L). The rate of recombination in females between the two X-linked loci, the target of X-shredding and the target of editing, has no significant effect because, under the baseline assumptions, females with a single copy of the edited allele are dominant lethal. Finally, population suppression is also robust to reasonable values for the intrinsic rate of population increase (Fig 4M). All these plots are for our baseline $R_m$ value of 6; additional sensitivity plots for alternative values of 2 and 12 are shown in S3 Fig.

## Population restriction using allele frequency differences

The ability of the double drive to increase in frequency from rare means that it will spread not only in the release population, but also in other populations into which there is even a small level of gene flow. This ability to spread across a landscape can contribute greatly to the efficiency of the release, but may also cause problems, as there may be non-target populations one would not want to impact. In principle, one way to restrict the spread of a gene drive is to exploit pre-existing sequence differences between target and non-target populations [16,26]. We now investigate how such targeting would work for our proposed design by considering the impact of different levels of pre-existing resistance.

As noted above, 100% editing resistance can still lead to the spread of the constructs and some suppression, so choosing a site for editing that is differentiated between target and non-target populations will not provide effective restriction. For the other two processes, homing and shredding, we simulated releases into populations with different levels of pre-existing resistance (example time courses shown in S4 Fig). Plots showing the maximum construct frequencies and minimum population sizes as a function of the frequency of pre-existing resistance for low and high EJR scenarios are shown in Fig 5. In each of the four cases analysed there is good suppression with no pre-existing resistance, and negligible suppression with 100% pre-existing resistance. For homing resistance in low EJR species the transition between these occurs sharply, with pre-existing resistance of 84% giving a minimum population size of 1e-6 (i.e., 99.9999% suppression), and pre-existing resistance of 87% giving a minimum population size of 0.95 (i.e., 5% suppression). However, suppression in the former case can take impractically long (hundreds of generations); if we specify that the target population must be

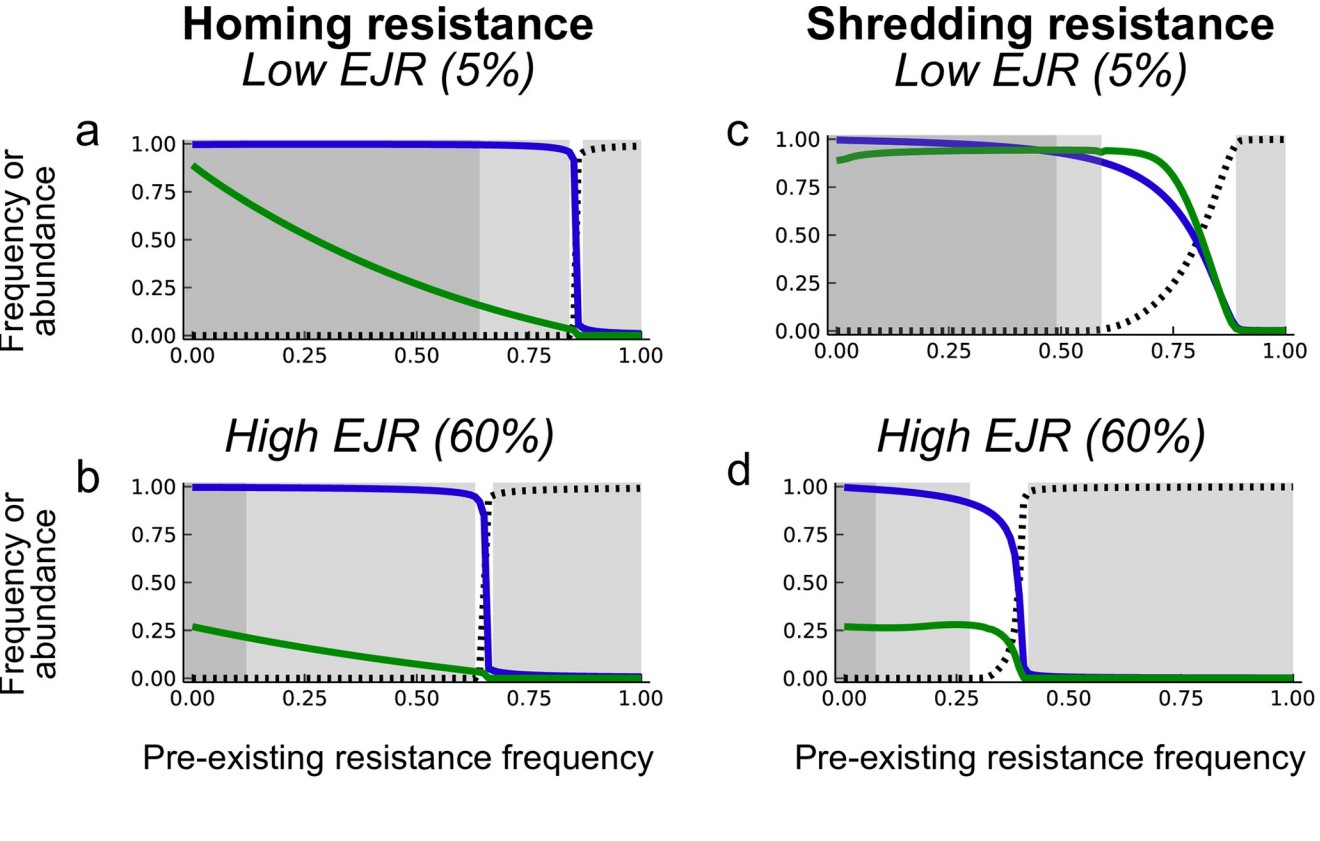

**Fig 5. Exploiting pre-existing sequence differences to restrict population impact.** Minimum population size and maximum construct frequencies as a function of the frequency of pre-existing resistance at the sites of homing and shredding for low and high EJR scenarios. Simulations were run for 1000 generations with different initial frequencies of resistant alleles. In each plot the lightly shaded boxes to the left and right show regions where the minimum population size over the 1000 generations is less than 1e-6 or greater than 0.95, respectively, as would be appropriate for target and non-target populations, and the darkly shaded box to the left shows regions where the population is suppressed by at least 99% in 50 generations, and therefore control may be practicable.

suppressed by 99% within 50 generations, then that requires the pre-existing resistant allele have a frequency less than 64%.

From an engineering point of view the fact that this transition from substantial to negligible control occurs in a relatively small window of pre-existing sequence differences (64–87%) is a good thing, as it reduces the level of overall population differentiation needed to find a suitably differentiated target site that would restrict the impact of a release to target populations. In the ideal case it would resemble a step function. The transition is less sharp for homing resistance in high EJR species (12–67%), and for shredding resistance in low and high EJR species the transition is intermediate between these extremes (Fig 5).

### PAM site analysis in *An. gambiae*

To better understand the potential opportunities for sequence-based population targeting that these results imply, we performed a preliminary analysis of published genome sequences from *An. gambiae* s.l. mosquitoes, analysing data from 15 populations and 1138 individuals across sub-Saharan Africa [18]. The most commonly used Cas9-based CRISPR nuclease recognises a protospacer adjacent motif (PAM) of NGG, so we screened the database for polymorphic GG (or CC) dinucleotides. After filtering out sites with more than 5% missing data within any

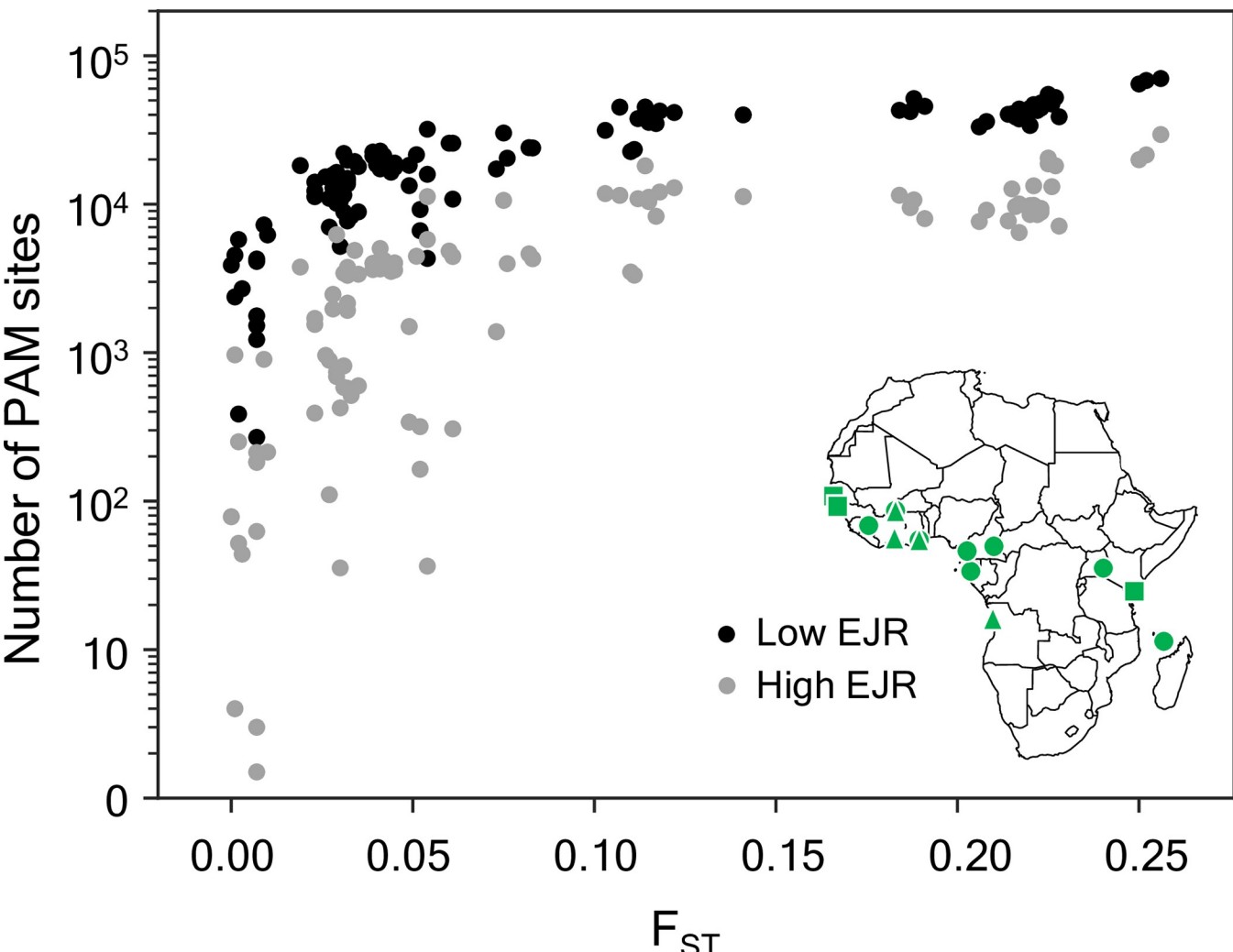

**Fig 6. Number of suitably differentiated PAM sites as a function of the overall differentiation between pairs of *An. gambiae* s.l. populations.** For each pair of populations we plot the average number of autosomal PAM sites with frequencies of >36% in one population and <13% in the other (low EJR scenario, black dots, equivalent to frequencies of resistant alleles of <64% and >87%, respectively) or >88% in one population and <33% in the other (high EJR scenario, grey dots). $F_{ST}$ values were obtained from The *Anopheles gambiae* 1000 Genomes Consortium ([18], Supp. Fig S5). Inset shows the location and subspecies assignment of the 15 populations (*An. gambiae* [circles]. *An. coluzzii* [triangles] or unassigned [squares]). The sample site map was generated using the cartopy python package (http://scitools.org.uk/cartopy).

population, we were left with 13,462,450 polymorphic PAMs. For each pair of populations we counted the number of PAM sites that had a frequency in one population >36% (the target population) and in the other <13% (non-target population), representing the level of differentiation required at the homing site in a low EJR species (Fig 5A), and then re-did the count in the opposite direction (i.e., reversing the assignments of target and non-target population). The average for the two directions was then plotted against the $F_{ST}$ for that pair of populations, the standard overall measure of population differentiation (Fig 6). As expected, the number of suitable PAM sites increased with $F_{ST}$, and for all population pairs with $F_{ST} > 0.02$ there were more than 4000 appropriately differentiated PAM sites. For less well differentiated populations there was substantial variation in the number of suitable PAM sites, from 200–8000 sites. We repeated the analysis using the thresholds for high EJR species (>88% and <33%), and, as expected, the number of suitable PAMs was lower, fewer than 10 in some instances, but most

population pairs had hundreds to tens of thousands. For extrapolating to other species, we note that the combined length of the autosomes in *An. gambiae* is about 206Mb, of which about 61% was accessible in these analyses.

## Discussion

Burt and Deredec [4] have demonstrated that YLEs could, in principle, provide highly efficient yet self-limiting population control, either on their own or augmented with an ASD. In principle, substantial suppression could be achieved even with a single release of less than 10% of the target population. However, for some potential use cases, even this number may be prohibitive, and more efficient strategies could be achieved with self-sustaining gene drive designs. In this paper we have proposed and analysed a method for getting the YLE to spread in a self-sustaining way using a double drive design, defined as "one that uses two constructs, inserted at different locations in the genome, both of which can increase in frequency, at least initially, and which interact such that the transmission of at least one of them depends on the other" [16]. The specific molecular adjustment made was to add a gRNA to the ASD, which means that not only does the ASD boost the transmission of the YLE, but the YLE also boosts the transmission of the ASD. This reciprocity allows both constructs to spread from rare, even if they are initially released in different males, and so they behave cooperatively to spread selfishly (from the rest of the genome's point of view). While there is reciprocity, the two constructs are not symmetrical and there is a division of labour between the YLE that is primarily responsible for imposing the reproductive load and the ASD that is responsible for the spread of the first. All our simulations assume a single release of 0.1% of the original population size, and, as with other self-spreading gene drive systems [27,28], we expect the main effect of having larger or repeated releases would be to reduce the time to impact. The ability of the constructs to spread and have an impact from arbitrarily small release rates is the key distinguishing feature of our double drive design compared to the self-limiting augmented YLE designs in which the ASD does not have a gRNA and cannot home [4].

Though we have analysed only a well-mixed non-spatial model, it is possible to make predictions about the dynamics and impact of a release across a landscape. The ability of the double drive to spread from rare, even if they are not initially in the same organisms, means that the constructs would be expected to spread geographically, to any population with which there is some reasonable level of gene flow. This is so despite a tendency for the intact ASD to eventually be lost within a population (Fig 2C and 2E). As long as the YLE and ASD are rare, the intact ASD is positively selected for, and as long as a few functional types reach a population, spread will ensue, so populations could be suppressed even far from the release site. That is, the X-shredder function should be evolutionarily stable enough to allow spread across an entire landscape. In principle, it may be possible to increase the evolutionary stability of the ASD by inserting it into a female-essential gene [22], as then mutants losing the X-shredder function would occur more often in females, and be selected against, though further modelling would be needed to confirm this idea, and our results suggest it may not be necessary. The editing function, on the other hand, though primarily responsible for the reproductive load imposed on the population, does not contribute to the spread of the constructs, and one would expect that eventually, if the species is sufficiently widespread (relative to dispersal distances), the gRNA component of the YLE will be lost. However, if the fitness cost of the gRNA in males is low, and the mutation rate (which only involves normal cell-division-associated DNA replication) is low, there could still be significant geographical suppression before that happened. This dynamic is analogous to a population replacement gene drive losing a cargo effector gene [29], but here the mutation rate may be much smaller (because the YLE does not home), so, all

else being equal, it should persist longer. In principle, it may be possible to engineer YLEs that will remain intact for different distances from the point of release by altering the mutation rate (e.g., adding repeated sequences to increase; adding redundant gRNAs to decrease) or fitness effects, though a reduced double drive will continue to spread beyond the point of the gRNA being lost, having a transient impact on population size due to the X-shredder. As in other population suppression gene drive systems, after the initial spread across a landscape there may be complex extinction-recolonisation or chasing dynamics [27,28,30,31], particularly in the absence of inbreeding depression or other strong Allee effects [32], but further investigation is beyond the scope of this study. Similarly, we have not considered the effects of finite population, overlapping generations, spatial or temporal heterogeneity, and other factors that one would want to include in a model tailored to predict more precisely the dynamics and impacts of a release in a specific potential use case. Different assumptions about the mode of action of the edited allele (e.g., if it affected embryonic survival or adult fertility instead of pupal emergence) would also have a quantitative effect on the dynamics of female numbers.

As with any proposed form of pest control, one needs to consider the possibility of resistance evolving. Because the reproductive load imposed on the population is primarily due to the edited locus, selection for target site resistance is strongest at that site, and the extent of population suppression is most sensitive to the probability of resistance arising at this site (Fig 3). Efforts to reduce the likelihood of resistance evolving should therefore be focussed on this site. This can be done by targeting key functional sites that are not able to change while maintaining function, and by targeting multiple sites in the same target gene [21,33,34], or even in different genes. By comparison, selection for resistance at the sites of homing and shredding is weaker, and higher rates of mutation to resistance can be tolerated.

Similar considerations apply to the required efficiencies of the different processes. Our sensitivity analysis shows that the requirements for the editing efficiency are the most stringent (>80% in the high EJR scenario with our baseline parameter values), while the requirements for the shredding and homing rates are lower. This difference arises because the YLE is insulated from the negative fitness effects it causes to daughters, and therefore selection against it is weak, and the processes increasing its frequency (shredding and homing) do not have to be very strong in order to get the YLE to a high frequency. Indeed, we have seen that the design can work even if most cleavage events at the autosomal site are repaired by end-joining. This robustness may be important because while high rates of homing and shredding have been achieved in anopheline mosquitoes, the rates observed thus far in some other species have been lower [12,14,35]. Thus our design may expand the range of species in which it is possible to engineer low release rate population suppression gene drives. We speculate that it may be possible to further reduce the required rates of homing and shredding if the X-shredding activity is conditional, only occurring in the presence of both constructs [4], though this may be more difficult to engineer if the sex chromosomes are silenced at meiosis. The design analysed here does not rely on expression of the sex chromosomes during meiosis, though it does require expression from the Y-chromosome in the germline. Such expression has been observed for a marker gene in *An. gambiae* [36] and for Cas9 in *Drosophila melanogaster* [37].

A particularly attractive feature of our design is the potential it offers for localisation of the spread and impact of the releases based on pre-existing sequence differences between target and non-target populations; these may be of the same species and occur in different locations, or they may be different sibling species in the same location (e.g., [38]). Such localisation can be achieved by choosing the insertion (and homing) site of the ASD or its target site on the X chromosome (or both) to have a suitably differentiated sequence. Willis and Burt [16] have previously shown that autosomal double drive designs may be able to work much better for population restriction than single construct designs, and with the design analysed here even

smaller differences in allele frequency can be exploited, as seen in the relatively steep change in outcome based on small differences in allele frequencies (Fig 5). This sensitive dependence of the dynamics on initial conditions (the frequency of pre-existing resistance) is reminiscent of that seen with so-called threshold-dependent gene drive designs, which spread if introduced above a particular frequency, and disappear if introduced below that frequency, and have been proposed as an alternative method to localise spread and impacts [39–43]. Our PAM site analysis of *An. gambiae* population genomic data indicates there may be very many suitably diverged target regions of the genome between even closely related populations. Such sequence-dependent localisation could be used to limit the spread and impact of a release either geographically or taxonomically, whenever the target taxon interbreeds to some extent with non-target taxa. That said, our analysis must be considered preliminary. For example, we have assumed that any difference in PAM site would be enough to block cleavage, but this assumption would need to be tested in the target species. In some systems the Cas9 from *Streptococcus pyogenes* can recognise not only the canonical NGG PAM, but also NAG, NGA, and NNGG, albeit with lower efficiency [44], and one might need to avoid these polymorphisms if the same occurs in the species of interest. We have not investigated whether these differentiated PAMs tend to be in coding or non-coding regions; if in the former, the ASD may need, for example, to be placed in an artificial intron [45] to reduce the fitness effects of gene disruption. We have also not investigated differentiation on the X chromosome, and it is unclear to which extent a shredding strategy would need to target a repeated sequence.

The final key feature of our design worth emphasising is that it is based on a small molecular modification of highly efficient self-limiting designs. Modern biotechnology has the potential to offer effective and sustainable forms of pest control that are safe for both people and the environment, but the very fact they are new has led to recommendations for a step-by-step approach in which increasingly efficacious constructs are assessed [46,47]. The starting point for our design is a genome editor that acts in males to produce dominant X-linked lethal or sterile mutations. Some of the molecular options for the editor are outlined in Burt and Deredec [4], including knocking out a haplo-insufficient gene, producing a dominant negative mutation, or using paternal deposition of the editor, and a proof-of-principle demonstration in *D. melanogaster* is provided by Fasulo et al [12]. Instead of using a cleavage-based editor like Cas9 it may be possible to use a base editor or prime editor [48,49], and instead of targeting an X-linked locus, it would also be possible to target one on an autosome, but the impact would have to be female-specific; potential targets are rare but do exist [16,50]. From this starting point, it is now possible to identify a step-wise progression of at least five designs of increasing efficacy:

1. An autosomal insertion of the construct, which would be transmitted 50% of the time to the daughters that are killed, and so halve in frequency each generation, and therefore have dynamics like a female-specific RIDL strain [51,52]

2. The same construct moved to the Y-chromosome (i.e., a YLE)

3. Augmenting the releases with an autosomal sex distorter to boost the frequency of the YLE

4. A localised sequence-specific YLE-based double drive (i.e., the design analysed in this paper)

5. A 'universal' (non-population-specific) YLE-based double drive

In principle, each of these steps could be further divided by first using constructs with high 'unintended' fitness costs, followed by constructs that have been optimised to minimise these costs. We are not suggesting that it will be necessary or even desirable to go through all steps

for all possible use cases. Rather, we merely wish to note that there is a great deal of flexibility in the designs, and therefore in the development pipeline, which can be optimised for any specific use case. This sort of step-by-step development pathway may prove attractive to developers, regulators, and other stakeholders more generally [53].

Sex chromosome systems vary widely among species, and in some species the repetitive and heterochromatic structure of the Y chromosome can make them difficult to work with. However, as we have seen, there can be considerable advantages to developing YLE-based interventions, including highly efficient self-limiting strategies; self-sustaining strategies that can be effective with lower rates of homologous repair and no requirement for meiotic expression off the sex chromosomes; more options for using pre-existing sequence differences to limit the spread and impact of a self-sustaining release; and a step-by-step development pipeline which may help reduce uncertainties and facilitate acceptance by developers, regulators, other stakeholders, and the relevant publics at large. Therefore, despite the difficulties of working with the Y chromosome, given the potential benefits of the approach, it seems well worth investing the time and effort needed to fully explore this approach. The recent rapid expansion of molecular engineering capabilities in multiple species [54,55] makes us optimistic the challenges can be met.

## Supporting information

**S1 File. Description of the model and PAM site analysis.**
(DOCX)

**S1 Fig.** Timecourse of gene and population dynamics when only one construct is released, either the YLE (a) or the ASD (b) for the idealised case of no mutation, no resistance, and no unintended fitness costs.
(DOCX)

**S2 Fig. Comparison of alternative population suppression designs using YLEs.** A YLE by itself can give good suppression with repeated 10% releases (a), but a single 10% release has little effect (b). A single 10% release of a YLE combined with a non-driving ASD can give good suppression (c), but a single release of 0.1% has little effect (d), whereas a 0.1% release of the double drive design considered in this paper can give good suppression (e). Note that the editing and shredding rates were set to maximum values (1) and released males are homozygous for the ASD in Burt & Deredec (2018), whereas they are slightly smaller in this publication (0.95 and 0.9, respectively) and the released males are heterozygous.
(DOCX)

**S3 Fig. Extended sensitivity analysis.** Each plot shows the minimum population size as a function of changes in the specified parameter assuming low EJR (5%; dotted lines) or high (60%; dashed lines) and intrinsic rate of increase of either 2 (orange), 6 (black) or 12 (purple). Note for (a) that the homing rate is equal to (cleavage rate) x (1-end joining rate); for (e) that the orange dotted line is below the y axis limit (1e-10); and for (m) that only 1 line is shown for each end-joining rate since the intrinsic rate of increase is the dependent variable on the x-axis.
(DOCX)

**S4 Fig. Timecourses for gene and population dynamics with different initial frequencies (0, 20, 40, 60, 80 or 100%) of either homing or shredding resistance in low and high EJR species.**
(DOCX)

## Acknowledgments

We thank John Connolly, Silke Fuchs and John Mumford for useful comments on a previous draft.

## Author Contributions

**Conceptualization:** Austin Burt.

**Data curation:** Katie Willis.

**Funding acquisition:** Austin Burt.

**Investigation:** René Geci, Katie Willis.

**Methodology:** René Geci, Katie Willis.

**Software:** René Geci, Katie Willis.

**Supervision:** Austin Burt.

**Validation:** René Geci, Austin Burt.

**Visualization:** René Geci, Katie Willis.

**Writing – original draft:** Austin Burt.

**Writing – review & editing:** René Geci, Katie Willis.

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
