## [Decision Letter · Decision Letter 0]

1 Sep 2022

Dear Dr Burt,

Thank you very much for submitting your Research Article entitled 'Gene drive designs for efficient and localisable population suppression using Y-linked editors' to PLOS Genetics.

The manuscript was fully evaluated at the editorial level and by independent peer reviewers. The reviewers appreciated the attention to an important problem, but raised some concerns about the current manuscript. Based on the reviews, we will not be able to accept this version of the manuscript, but we would be willing to review a revised version.

If you decide to revise the manuscript for further consideration at PLOS Genetics, please aim to resubmit within the next 60 days, unless it will take extra time to address the concerns of the reviewers, in which case we would appreciate an expected resubmission date by email to plosgenetics@plos.org.

[LINK]

Please do not hesitate to contact us if you have any concerns or questions.

Yours sincerely,

Jackson Champer

Academic Editor

PLOS Genetics

Gregory P. Copenhaver

Editor-in-Chief

PLOS Genetics

In general, both reviewers and myself are favorably disposed to the manuscript, and it can most likely be published with some revisions and additions (because of the likely need for data additions, I would classify this as a major revision). Please address all reviewer questions and comments in your response. I have a few additional comments.

If possible, it would be helpful to show genetic load at the time of peak population reduction to allow for easier cross-comparisons with other suppression drives. This is a more general property than reduction in population size, which is heavily dependent on the specific species, ecology, and model used.

In the discussion, the authors state that, “This can be done by targeting key functional sites that are not able to change while maintaining function, and by targeting multiple neighbouring sites in the same target gene (Champer et al. 2018; Kyrou et al. 2018; Champer et al. 2020). By comparison, selection for resistance at the sites of homing and shredding is weaker, and higher rates of mutation to resistance can be tolerated.”

In fact, the target sites in this gene need not be neighboring. This is a requirement for efficient homing drives with multiple gRNAs, but not for the toxin component of gene drives, for which end-joining repair (and thus more widely spaced gRNAs) is fine (Champer et al, BMC Biology, 2020). Even targeting different genes should be equally feasible in this situation.

This later point may be important if there are only 4000-8000 differentiable targeting sites. In this case, most genes won’t even have a single site. It might be necessary to have multiple possible targets, though even finding one haploinsufficient gene on the X-chromosome with a differentiable target might be very difficult. Unless I misunderstand this, perhaps a bit more skepticism for this method could be incorporated into the discussion. of course, I suppose not every target needs to be differentiable (especially the X-shredder), but the X-linked and probably the ASD gRNA target should be.

Another possible consideration is that the effect of mutations in the ASD component could perhaps be reduced if placed in a haplosufficient but essential female gene (much like the construct in Simoni et al, Nature Biotechnology, 2020). (The authors may have already intended this in which case ignore this comment). In this situation, if the construct loses function of its components (particularly the X-shredder) due to mutating, the allele would be less fit because it would be removed in females with combinations of ASD alleles. Normally, ASD alleles may not often be together because of the X-shredder component, which tends to keep the drive in male drive/wild-type heterozygotes.

Reviewer 2’s comment on sensitivity analysis is quite important to address, but it may be impractical to conduct a full sensitivity analysis on all possible combinations without bringing in some additional computational techniques and a large amount of additional data collection (due to the large number of parameters investigated - for an example see Sam Champer et al., PLOS Computational Biology, 2021). Perhaps the authors could use a combination of discussion together with a somewhat more limited set of new analysis or data to address this (focusing on the parameters deemed most likely to be important, most likely to be variable, and most likely to have interesting interactions with others).

Reviewer's Responses to Questions

**Comments to the Authors:**

Reviewer #1: In this manuscript, the authors describe and model a suppression gene drive that builds upon their previous work with Y-linked editors and autosomal sex distorters. They introduce a tweak by adding a gRNA that facilitates homing of the ASD element. The YLE is designed to create a dominant mutation on the X causing lethality or sterility in females, resulting in population suppression.

Although, as the authors point out, the molecular implementation might be difficult (expression from the Y), I really like the design. With its location on the Y, the main load inducing component in form of the editor is shielded from the negative effects it creates. When the YLE is present in an individual that also carries the shredder both elements get a boost in frequency. The YLE induces homing of the ASD and the ASD ensures that most of the offspring will inherit a Y and thus, the YLE.

The authors model population dynamics over a broad range of parameters, including LOF mutations of components, resistance at target sites, NHEJ events during homing, and fitness costs. They show that suppression works well for reasonable assumptions of these parameters. A single release with a low introduction percentage was used for all models. Thinking about a real world application, I would assume that multiple releases can be carried out, making suppression even stronger and working over a broader range of parameters as the modeling in the paper suggests.

Finally, the authors touch on the subject of drive confinement based on target site diversity between target and non-target populations and give an estimate on how many such sites can be identified in Anopheles populations.

I think this clever combination of existing mechanisms is a great addition to the ever expanding drive system toolbox.

Comments:

The paper is based on previous work by some of the authors on YLE with non homing ASD. From looking at the old work, the biggest difference seems to be in the case of separate releases of the elements with a constitutive active ASD which did not work well without a homing ASD. A plot comparing the old and new systems for a simple set of parameters could help emphasize the advantages of the YLE with a homing ASD.

Page 4 Methods: The authors say that a number of different molecular constructions could be used. For readers trying to build such systems it might be helpful if the authors could sketch out these different designs (maybe in a supplemental figure?).

Page 6: Is there an advantage in releasing the drive components in separate individuals? Does that help with the rearing of the strains? Why would one want to do that?

Page 7: 1 in 1000 incorrectly homed alleles seems low. Is this number inferred from the cited publications?

Page 9: What are the baseline values? The ones from page 7 or is everything else at 100% efficiency? Can the authors clarify?

Page 11: The authors perform a preliminary analysis to identify PAM sites that differ in frequency between target and non-target populations based on PAM sequence motif. There are additional restrictions that should be mentioned. In case of shredding, the PAM needs to be on the X (in multiple copies?). In the case of homing the PAM needs to be at a neutral site (not disrupting anything essential).

Figure 6: (high EJR scenario, blue dots). Should that be gray dots?

Is it possible to highlight the immediately neighboring populations in the plot, since those might be most relevant when trying to prevent drive from spilling over?

Page 14: The authors write...A particularly attractive feature of our design is the potential it offers for localization of the spread and impact of the releases based on preexisting sequence differences between target and non-target populations...

Wouldn't all (or most) sequence based drives have the potential for localization based on target sequence differences?

Reviewer #2: Geci et al. describe a double drive design that has a Y-linked editor (YLE) and an autosomal sex distorter (ASD), which help each other to 'drive'. YLE targets an essential gene on the X-chromromose (haplo-sufficient or haplo-insufficient) that cause lethality or sterility in the daughters. The editting takes place in the male germline, and does not rely on sex chromosome expression during meiosis, which is an advantage for some species. The ASD targets repeated sequences in the X chromosome (seperate from the YLE target) and shreds the chromosome during meiosis (which produces skewed sex ratio, driving the YLE). ASD can also home in the presence of the YLE in the germline, so YLE is 'driving' ASD. Neither contruct can 'drive' when the other is absent.

In order to investigate the spread of this double drive, Geci et al. develop a population genetic and population dynamic model (infinite population size) that is deterministic and panmictic (nonspatial) with nonoverlapping generations. These properties make it hard to extrapolate the results of the model to more realistic scenarios, however, it still make useful predictions that show this YLE-ASD double drive could suppress a population with low release rates and with no invasion threshold. They also show that the An. gambiae has thousands of suitable genomic sites that are differentiated even for weakly differentiated populations, which could restrict the spread of the drive to other non-target populations.

I would recommend Geci et al.'s work for publication with minor revision. I have three major comments: Firstly, I find the model description very short, I strongly suggest more detailed description of the model (e.g. in Willis and Burt, bioRxiv; and/or Burt and Deredec 2018). Secondly, even though Geci et al. carry out analyses for multiple scenarios (loss of function, evolution of resistance through nonhomologous end joining, fitness costs etc.), they do it with one at a time approach (local sensitivity analysis). Since they do not vary multiple factors at the same time, it is not possible to understand how the interaction among various parameters effect the results particularly if they interact nonlinearly. Are there truly no nonlinear interactions between any of the parameters presented? Either that should be convincingly presented, or a global sensitivity analysis should be carried out where parameters of interest are varied simultaneously. Lastly, given the assumptions of the model, more discussion should be done on the caveats of their model (panmictic, infinite population size, etc).

I also have a few minor comments below, which I hope the authors would find useful.

p.5. Could you provide more detail on the model code?

p.6. fig 2. I would recommend keeping the x axis range of a and b same as the others [0,150] for quick and easy visual comparison for all.

-Specify correlation in the legend as well (between YLE and ASD)

-It might be useful to include the sex ratio skew in the graphs as well? I wonder why the females not increasing after defective x-shredder increases around generation 100 in 2c? Probably hard to see because of the reduced population size, in that case sex ratio plotted on the same graph would be also be useful...

-Subtitle in c and d would be useful (e.g. “X-shredder” in c) like in other subfigures.

p.7. I am a bit confused about loss of function in X-shredder: “Despite the loss of the X-shredder, the population is still very substantially suppressed, because the YLE is still very frequent: the X-shredder functions primarily as a booster of the YLE, not as a source of reproductive load itself.” (also on Fig. 2c). Since YLE doesn't work by itself, later loss of X-shredder doesn't affect because by then YLE has reached really high proportions? (what is the critical high frequency, below which it will fail?) Could you explain a bit more? (sort of similar to Burt and Deredec 2018 where release ratios are higher with continuous releases for many generations).

p.10. Fig. 4. I would recommmend separeting the subfigures to different rows of relevant figures for easy visualization/interpretation (a-c), (d,e), (f-j), (k,l), and m.

-Why are figures 4c and 4d (almost?) identical? Could you explain?

p.12. Fig .6. should “sl” be “s.l/”?

blue dots should be light gray?

Supp. Fig. S5 in the reference?

The scale of the inset does not allow all 15 to be seen.

p.20. Supplementary Figure SF-2: third column title should be “low ejr (5%)”

fig. Caption “with” different initial

frequencies?

p.21. Supplementary information, the model: could you give more details on the model also the parameters used on mortality, currently only intrinsic rate of increase of the population is given in the supp. Table. Is theta density-dependent or independent?

You mention “7 X chromosome variants (Supplementary Figure SF-3)” but the figure has 6 and 1071 can be obtained by 6 X chromosome variants.

p. 25. Supplemental Table S-1: The range of values tested for sensitity analysis as well, rather than only the baseline values would be useful. (also as I have mentioned earlier, SA should be repeated with multiple parameters varied simultaneouly.)

**Have all data underlying the figures and results presented in the manuscript been provided?**

Reviewer #1: Yes

Reviewer #2: Yes

PLOS authors have the option to publish the peer review history of their article (what does this mean?). If published, this will include your full peer review and any attached files.

Reviewer #1: No

Reviewer #2: No

---

## [Editor Report · Decision Letter 1]

29 Nov 2022

Dear Dr Burt,

We are pleased to inform you that your manuscript entitled "Gene drive designs for efficient and localisable population suppression using Y-linked editors" has been editorially accepted for publication in PLOS Genetics. Congratulations!

Yours sincerely,

Jackson Champer

Academic Editor

PLOS Genetics

Gregory P. Copenhaver

Editor-in-Chief

PLOS Genetics

Comments from the reviewers (if applicable):

The authors’ revisions in response to reviewer comments are generally satisfactory, and the manuscript is now ready for publication. My only remaining comment is that the genetic load could also be shown for the situations in some panels of Figures 3-5 (either peak or longer-term, perhaps as a supplemental figure). This is optional, so I’ll leave it to the authors if they want to add this before the final proofs stage.

**Data Deposition**

http://datadryad.org/submit?journalID=pgenetics&manu=PGENETICS-D-22-00862R1

**Press Queries**

---

## [Editor Report · Acceptance letter]

22 Dec 2022

PGENETICS-D-22-00862R1 

Gene drive designs for efficient and localisable population suppression using Y-linked editors 

Dear Dr Burt, 

We are pleased to inform you that your manuscript entitled "Gene drive designs for efficient and localisable population suppression using Y-linked editors" has been formally accepted for publication in PLOS Genetics! Your manuscript is now with our production department and you will be notified of the publication date in due course.

With kind regards,

Anita Estes

PLOS Genetics

On behalf of:
